# Spatiotemporal Patterns in River Water Quality and Pollution Source Apportionment in the Arid Beichuan River Basin of Northwestern China Using Positive Matrix Factorization Receptor Modeling Techniques

**DOI:** 10.3390/ijerph17145015

**Published:** 2020-07-13

**Authors:** Lele Xiao, Qianqian Zhang, Chao Niu, Huiwei Wang

**Affiliations:** 1College of Geology and Environment, Xi’an University of Science and Technology, Xi’an 710054, China; xiaolele@xust.edu.cn (L.X.); niuchao@xust.edu.cn (C.N.); 2Hebei and China Geological Survey Key Laboratory of Groundwater Remediation, Institute of Hydrogeology and Environmental Geology, Chinese Academy of Geological Sciences, Shijiazhuang 050061, China; whuiwei@mail.cgs.gov.cn

**Keywords:** water quality, spatio-temporal patterns, pollution source, positive matrix factorization (PMF) model

## Abstract

Deteriorating surface water quality has become an important environmental problem in China. In this study, river water quality was monitored in July (wet season) and October (dry season) 2019 at 26 sites, and a water quality index (WQI) and positive matrix factorization (PMF) model were used to assess surface water quality and identify pollution sources in the Beichuan River basin, Qinghai Province, China. The results showed that 53.85% and 76.92% of TN, 11.54% and 34.62% of TP, 65.38% and 76.92% of Fe, and 11.54% and 15.38% of Mn samples in the dry and wet seasons, respectively, exceeded the Chinese Government’s Grade III standards for surface water quality. The spatial variation in water quality showed that it gradually deteriorated from upstream to downstream as a result of human activity. The temporal variation showed that water quality was poorer in the wet season than in the dry season because of the rainfall runoff effect. The PMF model outputs showed that the primary sources of pollution in the wet season were mineral weathering and organic pollution sources, domestic and industrial sewage, and agricultural and urban non-point pollution sources. However, in the dry season, the primary sources were mineral weathering and organic pollution sources, industrial sewage, and domestic sewage. Our results suggest that the point pollution sources (domestic and industrial sewage) should be more strictly controlled, as a priority, in order to prevent the continued deterioration in water quality.

## 1. Introduction

Surface water has always played a vital role in supplying fresh water for human consumption, agricultural and industrial needs, and recreational purposes [1]. In recent decades, rapid urbanization, industrialization and the growth in human populations around the world have all led to deterioration in the quality of surface water [2,3,4,5], which is of serious concern to regulators and scientists.

One critical step toward the effective control of surface water pollution is to identify the factors that affect water quality and pollution sources. In general, surface water quality in any region is controlled mainly by natural factors (rainfall, weather, basin physiography, soil erosion, etc.) and anthropogenic factors (domestic and industrial wastewater, agricultural activities, urban development, etc.) [6,7]. Su et al. [8] found that the water quality of the Qiantang River in eastern China was affected mainly by domestic sewage and agricultural pollution, industrial wastewater discharge, vehicle exhaust, sand mining, and mineral weathering. Njuguna et al. [9] recently reported that Mn, Cl and Al were major pollutants of the Tana River in Kenya and came from natural sources.

In order to accurately trace the sources of contamination in water, it is necessary to apply accurate methods of source analysis. The positive matrix factorization (PMF) model, is a new source analysis receptor model that can apply non-negative constraints to the factor load and score in the process of finding a solution. This allows it to optimize the results of the source analysis using the standard deviations of the data, which makes the factor load and score more interpretable and the physical meanings are clearer [10]. In recent years, the PMF model has been widely used in studies of the atmosphere [11,12] and soil [13,14] to identify potential pollution sources and to quantity the contributions of each source. However, applications of the PMF model to identify the pollution sources in water environment have rarely been reported.

The Beichuan River is a tributary of the Huangshui River, and flows through Xining City, Qinghai Province, China. It is the most important source of drinking water for Xining City, and its water quality is important for continued socioeconomic development and the safety of the drinking water for the local inhabitants. Due to rapid urbanization and industrialization, the river water quality has been seriously affected by human activity. A previous study found that the Huangshui River has been significantly impacted by industrial and domestic sewage, resulting in concentrations of ammonia nitrogen and total nitrogen exceeded the drinking water quality standard in China [15]. However, research on the spatio-temporal variation in water quality and a source analysis has not been carried out in the Beichuan River Basin; thus, it is difficult to implement a targeted restoration program for improving water quality.

The aims of this study were to: (1) determine the characteristics of the spatial and seasonal patterns of water quality of the Beichuan River; (2) evaluate the river water quality using a water quality index (WQI); and (3) identify the main pollution sources and quantify the spatial and seasonal variations in the proportional contributions of these sources by using the PMF model. The results could be helpful in understanding the characteristics of the pollution sources in the Beichuan River Basin and in developing effective water quality protection strategies.

## 2. Materials and Methods

### 2.1. Study Area

The Beichuan River Basin surrounds Xining City, Qinghai Province, Eastern China, and the Beichuan River is a first-class tributary of the Huangshui River and a second-class tributary of the Yellow River. It originates from the Daban Mountains in Datong County, and it flows from northwest to southeast into the Huangshui River at Xiaoqiao Town, Xining City. The Beichuan River is 154.2 km long and the river basin has an area of approximately 3371 km^2^. The climate of the region is the arid and semi-arid continental climate of the plateau, with a mean annual temperature 5.88 °C and a mean annual precipitation of 367.5 mm (most of the precipitation occurs between May and September). The major land uses in the watersheds are woodland, grassland, cultivated land, construction land, water body and wasteland.

### 2.2. Sample Collection and Analysis

Water samples were obtained from two field campaigns carried out in July 2019 (wet season) and October 2019 (dry season), from the upstream to the downstream of the Beichuan River. There were 26 sampling sites, including 9, 8, and 9 sites (BC01-BC09, BC10-BC17 and BC18-BC26) from the upstream, midstream and downstream sections of the river, respectively (Figure 1). River water samples were collected approximately 10–30 cm below the water surface. Twenty water chemical parameters were analyzed as follows: pH, dissolved oxygen (DO), total dissolved solids (TDS), potassium (K^+^), sodium (Na^+^), calcium (Ca^2+^), magnesium (Mg^2+^), nitrate (NO_3_^−^), nitrite (NO_2_^−^), ammonia (NH_3_^+^), chloride (Cl^−^), sulfate (SO_4_^2−^), total nitrogen (TN), total phosphorus (TP), chemical oxygen demand (COD), total organic carbon (TOC), aluminum (Al), iron (Fe), manganese (Mn), and lead (Pb).

The pH and DO values were measured in the field using a WTW Multi 340 i/SET multiparameter instrument (Germany). In accordance with the national Standard Method of China (GB3838-2002a), concentrations of the cations and metal in the water samples were measured using an inductively coupled plasma atomic emission spectrometer (Agilent 7500ce ICP-MS, Tokyo, Japan), while anion analyses were carried out using spectrophotometry (Perkin-Elmer Lambda 35, Waltham, MA, USA). TDS were measured using gravimetric methods, and COD was measured using alkaline permanganate oxidation. TOC was measured using a total carbon analyzer (Liquic TOC II, Elementar Analysensysteme GmbH, Hanau, Germany). TN was measured using the alkaline potassium persulfate digestion and UV spectrophotometric method, while TP was measured using the persulfate digestion spectrophotometric method. All analyses were carried out at the Groundwater Mineral Water and Environmental Monitoring Center at the Institute of Hydrogeology and Environmental Geology, Chinese Academy of Geological Sciences.

### 2.3. Data Analysis

#### 2.3.1. Positive Matrix Factorization (PMF) Model

PMF is a typical receptor model, that has been recommended by the U.S. Environmental Protection Agency (USEPA) as a general apportionment modeling tool [16]. It uses the correlation matrix and covariance matrix to simplify initial high-dimensional variables and it can be used without source profiles as inputs. The PMF model has one important advantage, that is, it weights the uncertainty of each data point and applies a non-negative constraint to the data, which ensures that the source contributions are always positive [2,17]. The PMF model has been widely used in recent years in source apportionment in atmospheric, soil and water environments [16,18]. The model can be expressed as follows:(1)$$x_{ij}=\overset{p}{\underset{k=1}{\sum }}g_{ik}f_{kj}+e_{ij}$$
where *x_ij_* is the concentration of the *j*th water quality parameter in the *i*th sample; *g_ik_* is the contribution of the *k*th source for *i* number of samples; and *f_kj_* is the concentration of the *j*th water quality parameter in the *k*th source. The residual error matrix *e_ij_* is obtained by minimizing the object function *Q*:(2)$$Q=\;\overset{n}{\underset{i=1}{\sum }}\;\overset{m}{\underset{j=1}{\sum }}{\left(\frac{e_{ij}}{\mu_{ij}}\right)}^{2}$$

In this equation, *μ_ij_* is the uncertainty in the *x_ij_* measurement, which is calculated from the method detection limit (MDL) and the standard deviations (SDs) of the surrogate standards. When the concentration of a water quality parameter is ≤MDL, the uncertainty is calculated as:(3)$$Unc=\frac{5}{6}\times MDL$$

Otherwise, it is calculated as:(4)$$Unc=\sqrt{{\left(\sigma +c\right)}^{2}+MDL^{2}}$$
where *σ* is the relative SD and *c* is the level of the river water quality parameter. The EPA PMF 5.0 model was used in this study.

In this study, concentration data (including 18 water quality parameters for 26 water samples) and uncertainty data files (including sampling and analytical errors) were used as the input data for the PMF model (EPA 5.0) to estimate the source contributions to water quality in the Beichuan River Basin. Because the PMF model exhibits rotational ambiguity, the number of factors and the Fpeak values need be tested many times for different initial seeds to determine the variability in the PMF analysis. Different values of the rotational parameter Fpeak (between −1.5 and +1.5, in steps of 0.1) were explored. When the factor was set as 3 and Fpeak as 0.5 for the wet and dry seasons, the runs of the PMF model were best (the robust Q value is lowest (300.85 and 471.40 for wet and dry seasons, respectively)) and passed the Bootstrap test.

#### 2.3.2. Water Quality Index (WQI)

The WQI is a vital indicator for assessing the quality of surface water and its suitability for drinking purposes [19,20]. The WQI was calculated by assigning a weight (*W_i_*) to each water quality indicator according to its relative importance in the overall quality of surface water for drinking purposes [21]. Water quality standards were mainly referred to the Grade III standard for surface water quality in China [22]. Where this standard lacked a given indicator, the World Health Organization (2011) [23] standards were referred to instead. The assigned weight (*W_i_*) and the relative weight (*RW_i_*) for each indicator are given in Table 1. The calculated WQI values were classified into five categories: excellent (WQI < 50), good (WQI = 50−100), poor (WQI = 100.1−200), very poor (WQI = 200.1−300), and unsuitable for human consumption (WQI > 300) [19].

The WQI was calculated as follows:(5)$$RW_{i}=\frac{W_{i}}{\sum_{i=1}^{n}W_{i}\;}$$
(6)$$Q_{i}=\frac{C_{i}}{S_{i}}\times 100$$
(7)$$SI_{i}=W_{i}\times Q_{i}$$
(8)$$WQI\;=\;\sum SI_{i}$$
where *Q_i_* is the quality rating, *C_i_* and *S_i_* represent the concentration (mg/L) and water quality standard of each water quality parameter, and *SI_i_* is the sub-index of the *i*th parameter.

## 3. Results and Discussion

### 3.1. Water Quality Parameters of the Beichuan River

A statistical summary of the Beichuan River water quality parameters from the 26 sampling sites in the dry and wet seasons is presented in Appendix A
Table A1. The results show that the pH was mildly alkaline (mean pH values were 8.62 and 8.51 in the dry and wet seasons, respectively), while DO varied in the range of 6.30–13.30 and 4.50–13.50 mg/L for the dry and wet seasons, respectively. The mean anion concentrations in the dry and wet seasons were in the following order: SO_4_^2−^ > Cl^−^ > NO_3_^−^ > NO_2_^−^, cations in the dry season were in the following order: Ca^2+^ > Na^+^ > Mg^2+^ > K^+^ > Fe > Al > NH_3_^+^ > Mn > Pb, while cations in the wet season were in the following order: Ca^2+^ > Na^+^ > Mg^2+^ > K^+^ > Fe > Al > Mn > NH_3_^+^ > Pb. The mean concentrations of TN and TP were 1.35 and 2.41 mg/L in the dry season, and 0.124 and 0.250 mg/L in the wet season, respectively. In the dry and wet season, 53.85% and 76.92% of TN samples and 11.54% and 34.62% of TP samples, respectively, exceeded the Grade III standards (1.0 mg/L for TN and 0.2 mg/L for TP) of the national quality standards for surface waters in China [22]. The Fe and Mn concentration in the river water had a mean value of 0.467 and 0.746 mg/L in the dry season, and 0.036 and 0.057 mg/L in the wet season, respectively. In the dry and wet season, 65.38% and 76.92% of Fe samples and 11.54% and 15.38% of Mn samples, respectively, exceeded the Grade III standards for surface water in China [22].

These results show that the Beichuan River is seriously polluted by nutrient-related contaminants (TN and TP) and metal ions (Fe and Mn), which indicates that its water quality is affected by anthropogenic activities [24,25], including domestic sewage and industrial wastewater discharge, use of agricultural fertilizers, and urban rainfall runoff. In addition, the concentrations of the TN, TP, Fe and Mn in the wet season were higher than in the dry season. This indicates that the rainfall runoff significantly affects the quality of river water in the wet season [9,26].

### 3.2. Water Quality Assessment of Beichuan River Using WQI

WQIs were calculated for the samples using the values of pH, DO, TDS, K^+^, Na^+^, Ca^2+^, Mg^2+^, SO_4_^2−^, Cl^−^, NO_3_^−^, NO_2_^−^, NH_3_^+^, TN, TP, COD, Fe, Mn, Al and Pb at the 26 sampling sites. The results for the calculated WQIs during the dry and wet seasons are given in Table 2. The calculated WQIs range from 32.49 to 118.14 and from 37.60 to 209.04 in the dry and wet seasons, respectively.

In the dry season, 34.62% of the river water samples were graded as excellent, 42.31% as good, and 23.07% as poor. In the wet season, however, 26.92% of the river water samples were graded as excellent, 34.92% as good, 34.92% as poor, and 3.84% as very poor. In general, the water quality of Beichuan River deteriorates in the wet season because the rainfall runoff washed surface contaminants into the river and its tributaries [26].

For the upstream sites, 88.88% of the samples were graded as excellent and 11.12% as good. However, the water quality deteriorates from upstream to downstream along the river, and 87.50% of the samples were graded as good and 12.50% as poor in the midstream sites. For the downstream sites, 22.22% of the samples were graded as good, 72.22% as poor, and 5.56% as very poor. One explanation for this effect is that the area upstream of the Beichuan River mainly consists of woodland and grassland, and thus the river water is less affected by human activity. In contrast, the midstream and downstream sections of the Beichuan River, located in Tatong county and Xining City are mainly cultivated and urban construction land, and thus the quality of the river water is affected by anthropogenic activities, such as domestic and industrial sewage, agricultural fertilizer, and urban non-point source pollution [27,28]. Therefore, the water quality in the upstream section is better than that in the midstream and downstream sections.

### 3.3. Spatio-Temporal Patterns in the River Water Quality

In this study, the pH, TDS, Cl^−^, TN, Fe, and TOC were selected in order to analyze the spatial and temporal patterns in the river water quality. As shown in Figure 2, the mean concentration of TDS, Cl^−^, TN, Fe, and TOC showed an increasing trend from upstream to downstream. This can be attributed to the effects of human activity on river water quality, which gradually deteriorated from upstream to downstream. The spatial pattern of pH was not obvious because it is affected by multiple factors [29].

Temporal variations in the river water quality are largely affected by natural processes (rainfall, weathering processes) and human activity (urban, industrial and agricultural activities) [30]. As shown in Figure 2, the mean concentrations of Cl^−^, TN, Fe, and TOC in the wet season were higher than those in the dry season, which is probably because these pollutants mostly come from agricultural and urban rainfall runoff [31,32]. However, the temporal trends for pH and TDS in the dry season were slightly higher than those in the wet season. This might be due to the intense evaporation that occurs in the dry season and dilution of excessive rainfall in the wet season [7].

### 3.4. Identifying the Main River Pollution Sources Based on the PMF Model

Three factors were identified in the wet and dry seasons based on the PMF model, and their concentrations are presented in Table 3. In the wet season, Factor 1 accounts for 50.06% of the sum of the measured water quality parameters and it has higher relative concentrations of TDS, K^+^, Na^+^, Ca^2+^, SO_4_^2−^, Cl^−^, COD and TOC. Sources of salt ions (K^+^, Na^+^, Ca^2+^, SO_4_^2−^ and Cl^−^) in the river could come from natural sources (oceans, atmospheric deposition, weathering of common rocks, and brines) and anthropogenic sources (domestic sewage, industrial wastewater, agricultural runoff, and application of deicing salts) [33]. In this study, the concentrations of these ions were relatively low, thus, they are not derived from anthropogenic sources. Koklu et al. [34] found that Na^+^, Ca^2+^ and SO_4_^2−^ in rivers may come from mineral weathering. Guo and Wang [35] also found that Ca may come from the dissolution of calcium-bearing minerals. Therefore, it was concluded that the ions in the Beichuan River mainly originate from mineral weathering. In addition, it is generally understood that COD and TOC are indicators of organic contaminants in water and represent organic pollution sources. Based on the above analysis, Factor 1 represents mineral weathering and organic pollution sources (natural source and organic pollution sources).

Factor 2 explains 31.67% of the sum of the measured water quality parameters and is dominated by TN, NO_3_^−^, Cl^−^, Mg^2+^, Al, Fe and Mn. TN and NO_3_^−^ represent nutrient pollution that arises from strong anthropogenic impacts such as domestic and industrial sewage, the use of fertilizers and animal manures, intensive agriculture, and elevated atmospheric N deposition [36]. Cl^−^ in river water can come from septic effluent, the dissolution of minerals, agricultural chemicals, and road salt [37]. In addition, researchers have found that high concentrations of Cl^−^ and NO_3_^−^ indicate the presence of sewage [38]. Indeed, during the sampling period, we found that domestic sewage from villages near the Beichuan River, especially in the midstream and downstream, was drained directly into the river without treatment. In addition, the concentration of Cl^−^ and NO_3_^−^ in the two samples of sewage was quite high in the study area (the mean concentration of Cl^−^ and NO_3_^−^ was 169.0 and 71.13 mg/L, respectively). Thus, the high concentrations of TN, NO_3_^−^ and Cl^−^ in the region may be derived from domestic sewage. Al, Fe and Mn represent pollution by metals and metallic compounds and likely originate from industrial wastewaters [39], most probably from steel processing plants located in Xining City (downstream of the Beichuan River). The mean concentration of Fe in the downstream was higher than that in the midstream and upstream sections of the river (Figure 2). It is worth noting that site 20 is located in an industrial park in Datong county, and the substantial discharge of industrial and domestic sewage into the Beichuan River has resulted in a high concentration of Cl^−^ (42.0 mg/L), TN (3.54 mg/L), Al (1.01 mg/L) and Fe (1.07 mg/L). Therefore, Factor 2 was selected to represent domestic and industrial sewage (point pollution source).

Factor 3 accounts for 18.27% of the sum of the measured water quality parameters and consists predominantly of TP and Pb. High concentrations of TP and TN in surface waters can come from domestic and industrial sewage, the application of fertilizer, and animal waste [4,40]. Factor 2 represents domestic and industrial sewage and each factor is independent of the others, therefore, TP and TN in Factor 3 may derive from agricultural non-point source pollution. As shown in Figure 1, there is a large area of farmland on both sides of the Beichuan River, and the residual fertilizers from the farmland are easily washed by rainfall runoff into the river. Indeed, at site 10 in the midstream there is a large area of agricultural land on both sides of the river (Figure 1). The fertilizer applied to the farmland is washed away by storm runoff and enter the Beichuan River, resulting in concentrations of TN and TP of 2.09 mg/L and 0.648 mg/L, respectively. High concentrations of Pb may result from traffic emissions [41] and wet and dry depositions from industrial activity. In this study, the mean concentration of Pb in the downstream (the mean value is 0.0031 mg/L, urban land use) is higher than that in the midstream (0.0019 mg/L, township land use) and upstream (0.0005 mg/L, forest land use), which suggests that the concentration of Pb increases gradually with a gradual increase in urban traffic. Previous studies have found that the concentrations of Pb are high in road runoff [42]. Thus, Pb in the Beichuan River may come from urban non-point pollution sources. Accordingly, Factor 3 was selected to represent agricultural and urban non-point pollution sources (non-point pollution source).

In the dry season, Factor 1 accounts for 52.20% of the sum of the measured water quality parameters and is consistent with mineral weathering and organic pollution sources (natural source and organic pollution source). It consists predominantly of TDS, K^+^, Na^+^, Ca^2+^, Mg^2+^, NO_3_^−^, SO_4_^2−^, Cl^−^, COD and TOC. Factor 2 explains 28.27% of the sum of the measured water quality parameters and is dominated by Al, Fe and Mn. Factor 2 represents industrial wastewater (point source). Factor 3 accounts for 19.53% of the sum of the measured water quality parameters and consists predominantly of Na^+^, Cl^−^, TN, TP and Pb. High concentrations of TN and TP in the surface water may come from domestic sewage due to much lower rainfall in the dry season. Thus, Factor 3 represents domestic sewage (point source).

### 3.5. Source Contribution Based on PMF Model

#### 3.5.1. Estimated Contribution (mg/L) of Each Source to 26 Sampling Sites

Figure 3 shows the average contribution (mg/L) for each source at 26 sampling sites based on the output of the PMF model. On the whole, the mean contribution of the mineral weathering and organic pollution sources to water quality parameters at 26 sampling sites in the dry season (306.1 mg/L) is higher than that in the wet season(274.3 mg/L), which is mainly due to the dilution of excessive rainfall in the wet season (Figure 3a). There was no obvious spatial variation in the contribution of mineral weathering and organic pollution sources at 26 sampling sites in the wet and dry season.

In the wet and dry season, the mean contribution of domestic and industrial sewage to water quality parameters at 26 sites was higher in the downstream (138.8 mg/L and 180.8 mg/L, respectively) than in the midstream (47.5 mg/L and 57.5 mg/L, respectively) and upstream (16.8 mg/L and 11.1 mg/L, respectively), which is primarily due to the intensity of human activity in the downstream area. It is worth noting that the contribution of domestic and industrial sewage to site 10 (40.6 mg/L and 44.8 mg/L, respectively) was higher in the wet and dry season. This could be because site 10 was located in an area of intensive human activity and there is a sewage outlet upstream of the site. Domestic sewage from nearby villages is discharged into the Beichuan River. In addition, there are several metal factories near site 10 and we found that the industrial wastewater is directly discharged into the Beichuan River. Site 19 is another important site that is affected by domestic and industrial sewage (the contribution of domestic and industrial sewage to site 19 was 99.3 mg/L and 185.4 mg/L in the wet and dry season, respectively). It is located downstream of the Datong county industrial Park (where the facilities include non-ferrous metal production, chemical raw materials and chemical product manufacturing, ferrous smelting, manufacture of building materials, etc.) and the industrial wastewater from some factories is illegally discharged into the Beichuan River. Therefore, the contribution of domestic and industrial sewage to site 10 and 19 is significantly enhanced.

The mean contribution of non-point pollution source to water quality parameters at 26 sites is also higher in the downstream area (7.6 mg/L) than in the midstream (4.5 mg/L) and upstream (1.3 mg/L). Indeed, in the upstream area, woodland and grassland are the main land use types and they are in a state of natural growth (without fertilization). Therefore, the contribution of non-point pollution source to water quality parameters at upstream sites was lower than at the midstream and downstream sites. However, in the midstream sites, especially near site 11, a large area of agricultural land is located on both sides of the river (Figure 1) and the fertilizers applied to the farmland are washed away by storm runoff and then enter the river. The highest contribution of non-point pollution source to water quality parameters was found at site 24 (19.2 mg/L). This was mainly because site 24 is located in Xining city, and it is influenced by various pollution sources from agricultural (fertilizers) and urban (household waste and heavy traffic losses) non-point sources.

#### 3.5.2. Estimated Contribution Rate (%) for Each Source to 18 Water Quality Variables

Figure 4 shows the “factor fingerprints” of 18 water quality variables in the wet and dry seasons. In the wet season, most water quality parameters are mainly affected by mineral weathering and organic pollution sources (100% of pH, 100% of DO, 80.61% of K^+^, 49.62% of Na^+^, 91.92% of Ca^2+^, 50.34% of Cl^−^, 82.12% of SO_4_^2−^, 78.19% of TDS, 93.51% of COD and 64.91% of TOC), domestic and industrial sewage (45.31% of TN, 66.36% of Mg^2+^, 52.24% of NO_3_^−^, 54.33% of Al, 100% of Mn and 66.10% of Fe) and non-point pollution sources (100% of TP and 100% of Pb).

In the dry season, point sources are the major sources of pollution of the Beichuan River. Most water quality parameters were affected mainly by mineral weathering and organic pollution sources (100% of pH, 90.87% of DO, 78.05% of K^+^, 46.05% of Na^+^, 92.32% of Ca^2+^, 75.08% of Mg^2+^, 77.02% of SO_4_^2−^, 50.76% of NO_3_^−^, 79.70% of TDS, 94.22% of COD and 97.88% of TOC), domestic sewage (58.24% of TN, 44.85% of Cl^−^,75.25% of TP and 100% of Pb) and industrial sewage (76.55% of Al, 100% of Mn and 77.42% of Fe).

Based on the results of the study, domestic sewage and industrial wastewater are still major sources of pollution in the Beichuan River, especially in the dry season (the contributing ratio was 47.8%). Therefore, it is necessary for local governments, with support from the national government, to fund and expand the capacity of sewage and wastewater treatment facilities, establish performance standards, and develop an enforcement strategy to ensure that the water quality of the river is protected, especially as a source of drinking water.

## 4. Conclusions

In this study, the WQI and PMF model were applied to elucidate the spatio-temporal patterns of river water quality and to identify the pollution sources in the wet and dry season in the Beichuan River Basin. The results showed that 53.85% and 76.92% of TN, 11.54% and 34.62% of TP, 65.38% and 76.92% of Fe, and 11.54% and 15.38% of Mn samples in the dry (July 2019) and wet (October 2019) seasons, respectively, exceeded the Grade III standards for surface water quality in China. As a result, the Beichuan River is seriously polluted by nutrient-related contaminants (TN and TP) and metal ions (Fe and Mn).

The WQI was used to assess river water quality, and the results showed that the quality gradually deteriorated from upstream to downstream as a result of human activity. River water quality was poorer in the wet season than in the dry season because the rainfall runoff washes surface contaminants into the river.

The spatial variations in river water quality showed that the concentrations of TDS, Cl^−^, TN, Fe, and TOC increased from upstream to downstream. The temporal variation in groundwater quality is affected by the rainfall runoff and as a result, the mean concentrations of Cl^−^, TN, Fe, and TOC were higher in the wet season than in the dry season.

The PMF model results showed that the primary sources of pollution in the wet season were mineral weathering and organic pollution sources (50.06%), followed by domestic and industrial sewage (31.67%), and agricultural and urban non-point pollution sources (18.27%). However, in the dry season, the primary sources were mineral weathering and organic pollution sources (52.20%), followed by industrial sewage (28.27%), and domestic sewage (19.53%).

According to the results of the study, domestic sewage and industrial wastewater are still major sources of pollution in the Beichuan River. Thus, it is necessary for local governments, with support from the National Government, to fund and expand the capacity of sewage and wastewater treatment facilities, establish performance standards, and develop an enforcement strategy to ensure that the water quality of the rivers is protected as a source of drinking water. In addition, although we have mainly discussed the variation in water quality in the Beichuan River in the dry season and the wet season in this study, we did not consider the temporal variability in water quality across years. Therefore, it is recommended that the inter-annual variation in Beichuan River water quality should be studied in the future.

## Figures and Tables

**Figure 1 ijerph-17-05015-f001:**
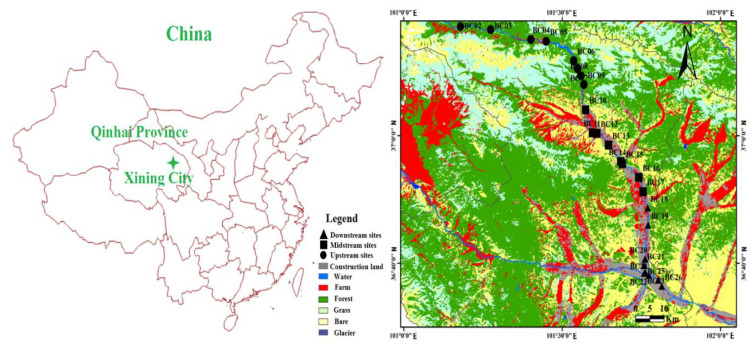
Water quality monitoring sites in the Beichuan River basin in China.

**Figure 2 ijerph-17-05015-f002:**
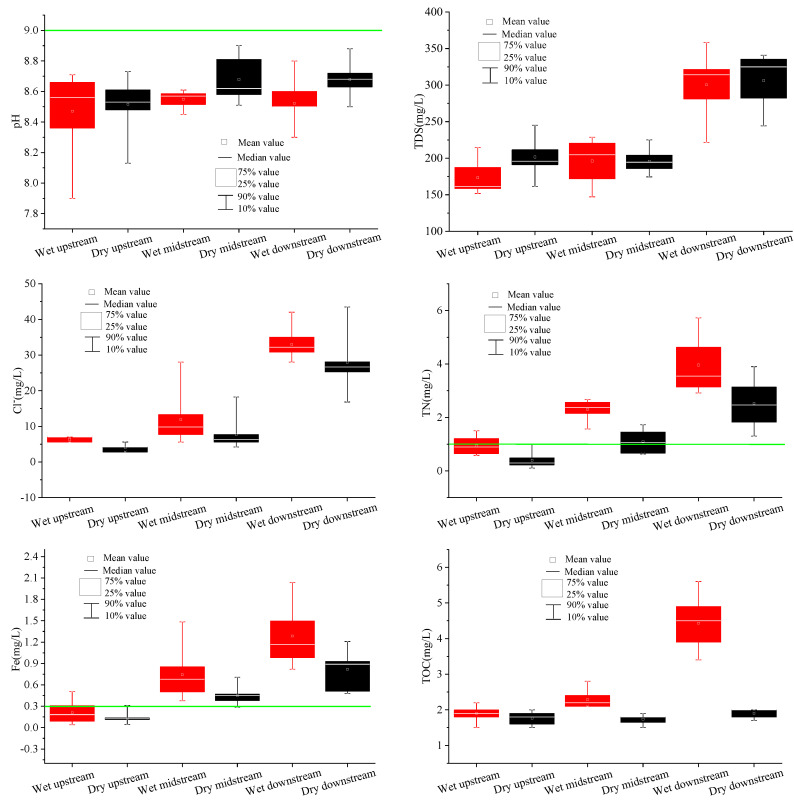
Temporal variations in six water parameters in the Beichuan River.

**Figure 3 ijerph-17-05015-f003:**
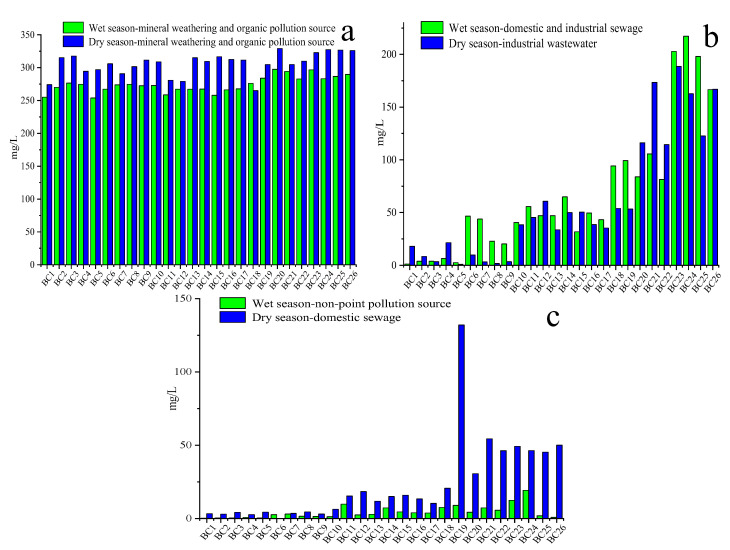
Estimated contributions (mg/L) from each source (mineral weathering and organic pollution source in the wet and dry seasons (**a**), domestic and industrial sewage in the wet season and industrial wastewater in the dry season (**b**), non-point pollution source in the wet season and domestic sewage in the dry season (**c**)) at the sampling sites obtained by the PMF model.

**Figure 4 ijerph-17-05015-f004:**
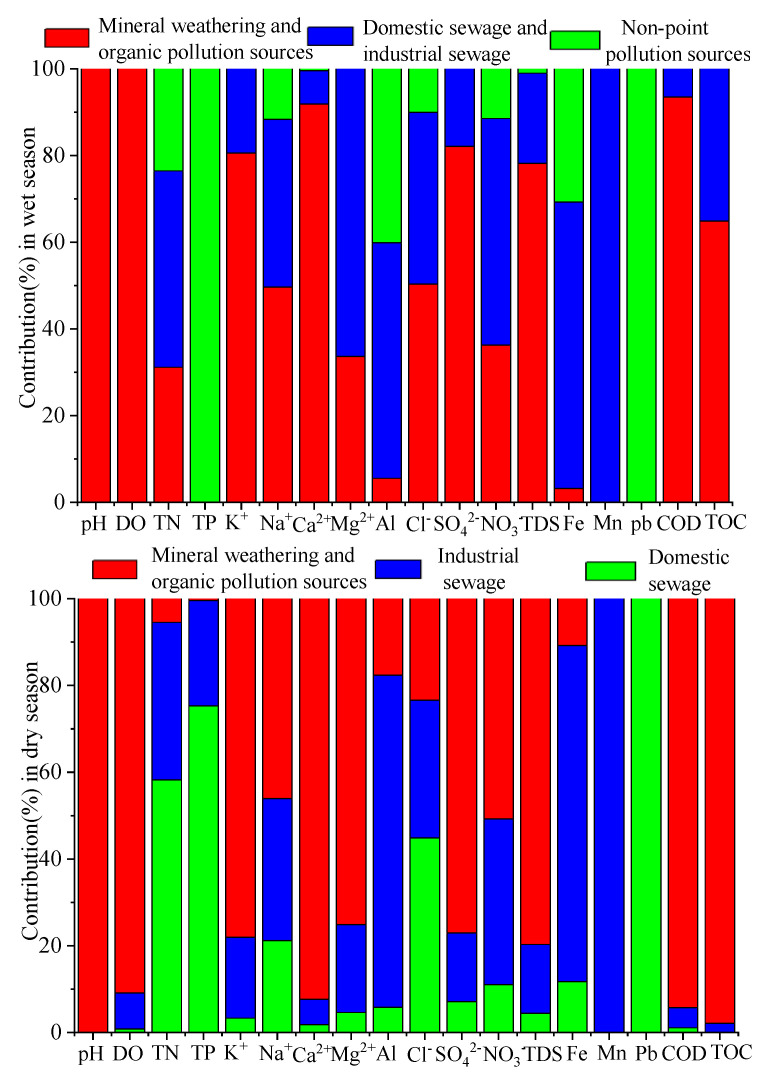
Factor figureprints of eighteen water quality variables resulted from the Environmental Protection Agency (EPA) PMF model in the wet and dry seasons.

**Table 1 ijerph-17-05015-t001:** Relative weight of water quality parameters and water quality standard (all units of the parameters are in mg/L except pH).

Parameters	Water Quality Standards	Weight (*Wi*)	Relative Weight (*RWi*)
pH	6.5–8.5	4	0.054
DO	5	4	0.054
TDS	1000	5	0.068
Na^+^	200	3	0.041
Ca^2+^	75	3	0.041
Mg^2+^	50	3	0.041
Cl^−^	250	5	0.068
SO_4_^2−^	250	5	0.068
TN	1	5	0.068
NH_3_^+^	1.22	5	0.068
NO_3_^−^	88.6	5	0.068
NO_2_^−^	3.29	5	0.068
TP	0.2	5	0.068
COD	3	5	0.068
Al	0.1	3	0.041
Fe	0.3	3	0.041
Mn	0.1	3	0.041
Pb	0.05	3	0.041
Sum		74	1

Note: The Mg^2+^ and Ca^2+^ parameters refer to the World Health Organization (2011) standards, the other parameters refer to the Grade III standard for surface water quality in China (GB3838-2002).

**Table 2 ijerph-17-05015-t002:** Water quality index (WQI) classification for dry and wet season in the Beichuan River.

Classification Type	WQI Values	Dry Season		Rainy Season	
		Number	Rate (%)	Number	Rate (%)
Excellent water	<50	9	34.62	7	26.92
Good water	50–100	11	42.31	9	34.62
Poor water	100.1–200	6	23.07	9	34.62
Very poor water	200.1–300	0	0	1	3.84
Water unsuitable for drinking purposes	>300	0	0	0	0

**Table 3 ijerph-17-05015-t003:** Source profiles obtained from the positive matrix factorization (PMF) model.

Parameters	Wet Season			Dry Season		
	Factor 1	Factor 2	Factor 3	Factor 1	Factor 2	Factor 3
pH	8.50	0.00	0.00	8.60	0.00	0.00
DO	7.23	0.00	0.00	7.43	0.68	0.07
TDS	164.47	43.81	2.06	183.23	36.45	10.21
K^+^	1.52	0.36	0.00	1.53	0.36	0.07
Na^+^	4.91	3.83	1.15	5.70	4.05	2.63
Ca^2+^	43.42	3.62	0.20	47.08	2.98	0.94
Mg^2+^	0.57	1.12	0.00	9.45	2.55	0.58
Cl^−^	5.15	4.05	1.02	2.32	3.14	4.44
SO_4_^2−^	31.44	6.84	0.00	33.79	6.95	3.14
TN	0.61	0.89	0.46	0.05	0.32	0.52
NO_3_^−^	2.79	4.02	0.88	3.32	2.50	0.72
TP	0.000	0.000	0.012	0.000	0.012	0.039
COD	1.64	0.11	0.00	1.61	0.08	0.02
TOC	1.74	0.94	0.00	1.75	0.04	0.00
Al	0.028	0.274	0.202	0.062	0.268	0.020
Fe	0.023	0.483	0.224	0.045	0.323	0.049
Mn	0.000	0.057	0.000	0.000	0.036	0.000
Pb	0.000	0.000	0.002	0.000	0.000	0.001
Possible sources	Mineral weathering and organic pollution source	Domestic and industrial sewage	Non-point pollution source	Mineral weathering and organic pollution source	Industrial wastewater	Domestic sewage
Contribution (%)	50.06	31.67	18.27	52.20	28.27	19.53

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
