# Peer review of "Spatiotemporal Patterns in River Water Quality and Pollution Source Apportionment in the Arid Beichuan River Basin of Northwestern China Using Positive Matrix Factorization Receptor Modeling Techniques"

_ijerph, 2020, doi:10.3390/ijerph17145015_

Round 1

Reviewer 1 Report

The manuscript has been modified according to the previous suggestions. Some minor changes can still contribute to its improvement.

Line 145: Replace "Calculated WQI values are usually classified..." with: "Calculated WQI values were classified..."

Figure 01. Change the color of the station points (not use green or red).

Table 2: remove the "Unit" column and mention in the caption that all units are in mg / L except pH

Author Response

Reviewer 1

The manuscript has been modified according to the previous suggestions. Some minor changes can still contribute to its improvement.

Response: Thanks for your suggestion.

1.Line 145: Replace "Calculated WQI values are usually classified..." with: "Calculated WQI values were classified..."

Response: We have replaced "Calculated WQI values are usually classified..." with "Calculated WQI values were classified..." in line 145.

2.Figure 01. Change the color of the station points (not use green or red).

Response: We have changed the color of the station points as black.

3.Table 2: remove the "Unit" column and mention in the caption that all units are in mg / L except pH

Response: We have advised the Table 2 according to your suggestions.

Reviewer 2 Report

  1. A very good study using PMF to identify sources of pollution in Northern China when of in-stream water quality is a concern as a Drinking Water source. Also the study was conducted as a partnership between government and universities to bring Science into Public Policy decision making.
  2. In the Discussion or Conclusion Sections, please indicate limitations such as, Study was for one year 2019, and does not account for temporal variability across Years (unless there is data to support that the variation across years is minimal).
  3. Pg 3 Lines 97 - 108; Please cite Standard Methods Reference Manual for the Methodology, such as China, EU or the US. While the Instrumentation manufacturer is important to cite, that alone does not assure standardized testing methods to ensure test results reliability.
  4. Please state the WQS Measurement Units in Table 1 and adjust to a single page, unless maintaining Headers when the page is split.
  5. Pg 5, Line 173 Full sentence, 'over standard rate of samples ..' is not clear. Water Quality (WQ) and impacts are both a function of concentration and volume. In wet weather, flow rates and volumes, and duration significantly affect WQ as related to Drinking Water Use.
  6. Pg 6, Line 185, rewrite as, '...WQ deteriorates in the wet season because' Need to say it is (for anthropogenic contributions) because of rainfall runoff effects, to differentiate wet and dry weather.
  7. Consider moving Table 2 to an Appendix Section because it is informative but with much data, that is not central to the Results.
  8. Pg 9, Line 230, Sentence appears incomplete unless linked to Factors - 'COD and TOC..'
  9. Table 4 should be a single page when formatted, unless Headers are repeated when on separate pages.
  10. Pg 10, Line 264. The explanation for sources of Pb in the river is not compelling. While roadway runoff is one source, so can be wet and dry deposition from industrial activity (unless there is study for the area that excludes this source).
  11. Pg 11, Line 276, Delete 'Thus'; Line 291, Replace 'not' with 'no'.
  12. Pg 13, Line 340 - 344. Rewrite paragraph to say, 'Based on the Results of the Study, domestic sewage and industrial wastewater are still major...' to keep the accepted terms consistent. Move Para on Line 342 to the Conclusion Section (repetitive at present, See Pg 14 Line 365) and make the recommendation to Government Agencies clearer -- e.g., 'It is necessary for local governments, with support from the National Government, to fund and expand the capacity of sewage and wastewater treatment facilities, establish performance standards, and develop an enforcement strategy to ensure the water quality of the rivers as a source of drinking water are protected'
  13. Pg 14. Ln 354, Consider clarifying sentence because, only concentration based measure is better because of dilution, but wet weather flows add to dry weather flows and dilution makes the in-stream WQ look better. This is not true until one can verify impacts. Short term exceedences when it rains of WQ in stream, does not necessarily mean the water is not good for drinking, unless one analyses persistence of the exceedance and its impact.
  14. Page 14, Ln 380, Rewrite as, 'The authors gratefully acknowledge ..the financial support from the public agencies listed above.'

Author Response

Reviewer 2

Comments and Suggestions for Authors

  1. A very good study using PMF to identify sources of pollution in Northern China when of in-stream water quality is a concern as a Drinking Water source. Also the study was conducted as a partnership between government and universities to bring Science into Public Policy decision making.

Response: Thanks for your suggestion.

  1. In the Discussion or Conclusion Sections, please indicate limitations such as, Study was for one year 2019, and does not account for temporal variability across Years (unless there is data to support that the variation across years is minimal).

Response: Thanks for your valuable suggestion. We have added the research limitations in the Conclusion Sections in line 391-395.

  1. Pg 3 Lines 97 - 108; Please cite Standard Methods Reference Manual for the Methodology, such as China, EU or the US. While the Instrumentation manufacturer is important to cite, that alone does not assure standardized testing methods to ensure test results reliability.

Response: Thanks for your valuable suggestion. We have added the standardized testing methods in line 98-99.

  1. Please state the WQS Measurement Units in Table 1 and adjust to a single page, unless maintaining Headers when the page is split.

Response: We have added the WQS Measurement Units in Table 1 and adjusted the Table 1 to a single page.

  1. Pg 5, Line 173 Full sentence, 'over standard rate of samples ..' is not clear. Water Quality (WQ) and impacts are both a function of concentration and volume. In wet weather, flow rates and volumes, and duration significantly affect WQ as related to Drinking Water Use.

Response: We have revised “In addition, the concentrations and over standard rate of samples for the TN, TP, Fe and Mn in wet season was higher than in the dry season.” as “In addition, the concentrations of TN, TP, Fe and Mn in wet season was higher than in the dry season” in line 182-183.

  1. Pg 6, Line 185, rewrite as, '...WQ deteriorates in the wet season because' Need to say it is (for anthropogenic contributions) because of rainfall runoff effects, to differentiate wet and dry weather.

Response: We have revised “In general, the quality of Beichuan River water was deteriorated in the wet season, which could have been due to the rainfall runoff washing surface contaminants into the river and its tributaries” as “In general, the water quality of Beichuan River was deteriorated in the wet season because the rainfall runoff washed surface contaminants into the river and its tributaries” in line 192-194.

  1. Consider moving Table 2 to an Appendix Section because it is informative but with much data, that is not central to the Results.

Response: We have moved Table 2 to an Appendix Section.

  1. Pg 9, Line 230, Sentence appears incomplete unless linked to Factors - 'COD and TOC..'

Response: We have revised “Therefore, it was concluded that the ions originated mainly from mineral weathering. COD and TOC are the indicators of organic contaminants in water and represent organic pollution source.” as “Therefore, it was concluded that the ions in Beichuan River originated mainly from mineral weathering. In addition, it is generally believed that COD and TOC are the indicators of organic contaminants in water and represent organic pollution source.” in line 236-238.

  1. Table 4 should be a single page when formatted, unless Headers are repeated when on separate pages.

Response: We have moved Table 4 to a single page.

  1. Pg 10, Line 264. The explanation for sources of Pb in the river is not compelling. While roadway runoff is one source, so can be wet and dry deposition from industrial activity (unless there is study for the area that excludes this source).

Response: Thanks for your valuable suggestion. We have revised “High concentrations of Pb may result from traffic emission sources [41]” as “High concentrations of Pb may result from traffic emission [41] and wet and dry deposition from industrial activity.” Due to both the traffic emission and wet and dry deposition from industrial activity enter the river driven by rainfall runoff. Therefore, we thought that Pb in the Beichuan River may come from the urban non-point pollution sources.

  1. Pg 11, Line 276, Delete 'Thus'; Line 291, Replace 'not' with 'no'.

Response: We have deleted the “Thus” in Line 276 and have replaced “not”

with “no”.

  1. Pg 13, Line 340 - 344. Rewrite paragraph to say, 'Based on the Results of the Study, domestic sewage and industrial wastewater are still major...' to keep the accepted terms consistent. Move Para on Line 342 to the Conclusion Section (repetitive at present, See Pg 14 Line 365) and make the recommendation to Government Agencies clearer -- e.g., 'It is necessary for local governments, with support from the National Government, to fund and expand the capacity of sewage and wastewater treatment facilities, establish performance standards, and develop an enforcement strategy to ensure the water quality of the rivers as a source of drinking water are protected'

Response: Thanks for your valuable suggestion. We have revised “Based on the above discussion, domestic and industrial sewage is still a major source of pollution in the Beichuan River,” as “Based on the results of the study, domestic sewage and industrial wastewater are still major source of pollution in the Beichuan River”in line 360-361, and revised “it is necessary for the local government to strengthen the capacity of sewage treatment and strictly control discharge of untreated sewage in order to prevent the continued deterioration of water quality” as “it is necessary for local governments, with support from the national government, to fund and expand the capacity of sewage and wastewater treatment facilities, establish performance standards, and develop an enforcement strategy to ensure that the water quality of the rivers as a source of drinking water is protected” in line 362-365, and moved the sentence to the Conclusion Section.

  1. Pg 14. Ln 354, Consider clarifying sentence because, only concentration based measure is better because of dilution, but wet weather flows add to dry weather flows and dilution makes the in-stream WQ look better. This is not true until one can verify impacts. Short term exceedences when it rains of WQ in stream, does not necessarily mean the water is not good for drinking, unless one analyses persistence of the exceedance and its impact.

Response: Thanks for your valuable suggestion. We have advised “River water quality was poorer in the wet season than in the dry season because of the rainfall runoff effect” as “River water quality was poorer in the wet season than in the dry season because of the rainfall runoff washed surface contaminants into the river.” in line 375-377. In this study, the water quality of the Beichuan River was better in the dry season than in the wet season. This mainly due to the surface pollutants (including agriculture fertilizers, domestic and industrial waste, heavy traffic losses and dry deposition) was washed by rainfall runoff into river. Therefore, the water quality of the Beichuan River was deteriorated in the dry season.   

  1. Page 14, Ln 380, Rewrite as, 'The authors gratefully acknowledge ..the financial support from the public agencies listed above.'

Response: We have rewritten the sentence according to your suggestion.

This manuscript is a resubmission of an earlier submission. The following is a list of the peer review reports and author responses from that submission.

Round 1

Reviewer 1 Report

The manuscript deals with a relevant topic and is within the scope of the journal. However, improvements are needed.

Figure 1: authors must include geographical coordinates at the edges of the map. The legend table and the country map must be placed outside the main map. The color or shape of the points that represent the monitoring stations can be changed to correspond to the “upstream”, “midstream” and “downstream” sections.

Item 2.3.2. Were the weights used in the water quality index chosen by the authors or based on other works? Make it clear in the text, if necessary include the quote.

Item 3.2 Line 168. The phrase “Calculated WQI values ​​are usually classified into five categories: excellent ...” should be transferred to item 2.3.2.

Table 2. In caption: mg / l is not a pH unit.
The presentation of the data in the table would be better if the data of the dry and rainy season were arranged side by side. Ex. Mean Dry | Mean Wet; SD Dry | SD Wet, etc.
.
Item 3.3. Line 213. Transfer the excerpt: “In this study, the EPA ... improve oblique edges” to item 2.3.1.

Figure 2: Increase the text size on the labels. To improve the visualization of trends and seasonality, use only two colors in the graphs. Ex: Black for all Wet station boxes and Red for all Dry station boxes. Remove the caption of the explanation from the boxplot and insert the information textually in the caption of the figure.

Item 3.5: Improve the discussion, related to the results found to the results of other similar studies. Also use quotes to support hypotheses.

Figure 3. Increase the text size on labels and captions. Use the same color, in both graphics, for the class “Mineral weathering and organic pollution source”.

Author Response

Reviewer #1:

Comments and Suggestions for Authors

The manuscript deals with a relevant topic and is within the scope of the journal. However, improvements are needed.

Figure 1: authors must include geographical coordinates at the edges of the map. The legend table and the country map must be placed outside the main map. The color or shape of the points that represent the monitoring stations can be changed to correspond to the “upstream”, “midstream” and “downstream” sections.

Response: We have added geographical coordinates at the edges of the map and have moved the legend table and the country map to outside the main map. We have changed the shape of the points that represent the monitoring stations to correspond to the “upstream”, “midstream” and “downstream” sections.

Item 2.3.2. Were the weights used in the water quality index chosen by the authors or based on other works? Make it clear in the text, if necessary include the quote.

Response: The weight to each water quality indicator were assigned according to its relative importance in the overall quality of surface water for drinking purposes and also referred to other work. We have explained it and added the reference in line 140-141.

Item 3.2 Line 168. The phrase “Calculated WQI values ​​are usually classified into five categories: excellent ...” should be transferred to item 2.3.2.

Response: We have transferred the phrase “Calculated WQI values ​​are usually classified into five categories: excellent ...” to item 2.3.2 in line 145-147.

Table 2. In caption: mg / l is not a pH unit.
The presentation of the data in the table would be better if the data of the dry and rainy season were arranged side by side. Ex. Mean Dry | Mean Wet; SD Dry | SD Wet, etc.

Response: Thanks for your suggestion. We have revised the Table 1 accordingly to your suggestion.

.
Item 3.3. Line 213. Transfer the excerpt: “In this study, the EPA ... improve oblique edges” to item 2.3.1.

Response: We have transferred the excerpt: “In this study, the EPA ... improve oblique edges” to item 2.3.1.

Figure 2: Increase the text size on the labels. To improve the visualization of trends and seasonality, use only two colors in the graphs. Ex: Black for all Wet station boxes and Red for all Dry station boxes. Remove the caption of the explanation from the boxplot and insert the information textually in the caption of the figure.

Response: We have added the text size on the labels and changed the colors of boxes according to your suggestion. In addition, we have removed the caption of the explanation and insert in the figure.

Item 3.5: Improve the discussion, related to the results found to the results of other similar studies. Also use quotes to support hypotheses.

Response: We have added some additional information on pollution sources within the catchment and quoted some references to support our conclusions in line 193-200 , 227-229 and 240-243, and so on.

Figure 3. Increase the text size on labels and captions. Use the same color, in both graphics, for the class “Mineral weathering and organic pollution source”.

Response: We have revised the text size on labels and captions and used the same color for same pollution source in Figure 3.

Reviewer 2 Report

The authors have applied the PMF model to water quality data collected from the Beichuan River, Qinghai Province, China. They sample at 26 sites, once in the dry season and once in the wet season. They analyse for 18 water quality variables, presenting tables of the values they find. The PMF model is a constrained weighted factor analysis which seeks to identify a small number of factors which explain most of the variability in the multivariate water quality data set which the authors have available. The authors identify three factors, one set of 3 for the dry season and one set for the wet season. They then interpret these factors as representing particular sources of water.

The tabulated values of water quality in the Beichuan River are useful in classifying water quality at the times and locations selected. The application of the model appears satisfactory, and the paper is well-structured and written in acceptable English. A single sample is taken at each site in the dry season and a single sample in the wet season. The reader therefore has no idea of the variability in water quality within either of these seasons, or from year to year. The factors identified are mathematical constructs, and the authors’ interpretation of them as representing particular source types is open to debate. Much of the authors’ discussion on this is speculative, but there must be a good deal of additional information on sources within the catchment, such as the location and likely composition of point discharges to the river. The authors do not present the gik values, which would presumably show trends from upstream to downstream and help identify locations at which water quality changed.

The script I have received contains author instructions after the Conclusion, and no reference list.

Figures

Figure 1: Cannot distinguish farm from forest. They are both a very similar green. What is “County”? Urban?

Figure 2: Increase the size of the axis labelling. How have the variables shown been

Author Response

Reviewer 2

Comments and Suggestions for Authors

The authors have applied the PMF model to water quality data collected from the Beichuan River, Qinghai Province, China. They sample at 26 sites, once in the dry season and once in the wet season. They analyse for 18 water quality variables, presenting tables of the values they find. The PMF model is a constrained weighted factor analysis which seeks to identify a small number of factors which explain most of the variability in the multivariate water quality data set which the authors have available. The authors identify three factors, one set of 3 for the dry season and one set for the wet season. They then interpret these factors as representing particular sources of water.

The tabulated values of water quality in the Beichuan River are useful in classifying water quality at the times and locations selected. The application of the model appears satisfactory, and the paper is well-structured and written in acceptable English. A single sample is taken at each site in the dry season and a single sample in the wet season. The reader therefore has no idea of the variability in water quality within either of these seasons, or from year to year.

Response: Thanks for your suggestion. Based on previous researches, the temporal variation of river water quality is primarily controlled by the monsoon (precipitation).  Therefore, in this study, we mainly discussed the variation of water quality and pollution sources between dry season and wet season in the Beichuan River basin.

The factors identified are mathematical constructs, and the authors’ interpretation of them as representing particular source types is open to debate. Much of the authors’ discussion on this is speculative, but there must be a good deal of additional information on sources within the catchment, such as the location and likely composition of point discharges to the river.

Response: Thanks for your suggestion. We have added some additional information on sources within the catchment in line 240-242 and 255-257.

The authors do not present the gik values, which would presumably show trends from upstream to downstream and help identify locations at which water quality changed.

The script I have received contains author instructions after the Conclusion, and no reference list.

Response: Sorry, I don't understand the meaning of GIK value.

Figures

Figure 1: Cannot distinguish farm from forest. They are both a very similar green. What is “County”? Urban?

Response: We have revised the color of the farm land and changed "county" as "Construction land".

Figure 2: Increase the size of the axis labelling. How have the variables shown been

Response: We have increased the size of the axis labelling and changed the colors of boxes according to different seasons.

Reviewer 3 Report

The report was interesting especially with the application of the PMF Model to water quality analysis. While the WQI index provided the river’s status, the PMF Model provided insight into the spatial and temporal variations.

This article will benefit by providing some clarity on other urban components of the river and on the geology. For example, we learnt about the major land uses on lines 82-83 and, read about the source of pollution by  mineral weathering at the discussion. It was not clear how the PMF Model was used to describe the main sources of pollution at the results/discussion section.   

Provide further information on how the PMF Model was applied to the water variables.

Lines 174-175 -----In general, the quality of Beichuan River water was deteriorated in the wet season, which could have been due to the rainfall runoff washing surface contaminants into the river and its tributaries

--------We learnt that the TDS reduced during the wet season. Could it be that the nutrients were not diluting during the wet season? What is the dilution rate during the wet season compared to the dry season in the river? 

Line 43: delete the comma after the reference ----Su et al

Lines 180-185: More information is needed to clearly describe the changes between the mid and downstream sections of the river. What form of activities were taking place there?

Line 187—delete ……..According to the characteristics of…

Lines 187, 189: Be consistent in the use of ionic charges. On lines 187,189 Cl had a ion symbol, but not Fe.

Line 220—delete--- figure illustrates

Lines 213-218--- more clarity is required here

Lines 224-225, It was not clear how this conclusion was decided. However, it would be good to focus the results in view of the PMF Model outcomes. 

Author Response

Reviewer 3

Comments and Suggestions for Authors

The report was interesting especially with the application of the PMF Model to water quality analysis. While the WQI index provided the river’s status, the PMF Model provided insight into the spatial and temporal variations. This article will benefit by providing some clarity on other urban components of the river and on the geology. For example, we learnt about the major land uses on lines 82-83 and, read about the source of pollution by mineral weathering at the discussion.

It was not clear how the PMF Model was used to describe the main sources of pollution at the results/discussion section. Provide further information on how the PMF Model was applied to the water variables.

Response: We have added the explanation of how the PMF Model was applied to the water variables in line 129-137.

Lines 174-175 -----In general, the quality of Beichuan River water was deteriorated in the wet season, which could have been due to the rainfall runoff washing surface contaminants into the river and its tributaries

--------We learnt that the TDS reduced during the wet season. Could it be that the nutrients were not diluting during the wet season? What is the dilution rate during the wet season compared to the dry season in the river? 

Response: Indeed, in this study, the temporal trends of TDS in the dry season was slightly higher than in the wet season, however, the concentration of nutrients(such as TN) was higher in wet season than in dry season. This could due to TDS mainly come from the weathering of minerals and the concentration of TDS in rainfall is low, thus,  TDS in river could be diluted by rainfall. However, TN in river may be derived from human activities (such as, agricultural fertilizers, domestic sewage and manure), and rainfall runoff washed surface contaminants(agriculture fertilizer and manure, etc) into the river in wet season. Therefore, the concentration of nutrients was higher in wet season than in dry season.

Line 43: delete the comma after the reference ----Su et al

Response: We have deleted the comma after the reference in line 43.

Lines 180-185: More information is needed to clearly describe the changes between the mid and downstream sections of the river. What form of activities were taking place there?

Response: We have described the changes between the mid and downstream sections of the river in line 194-200.

Line 187—delete ……..According to the characteristics of…

Response: We have deleted the sentence.

Lines 187, 189: Be consistent in the use of ionic charges. On lines 187,189 Cl had a ion symbol, but not Fe.

Response: Thanks for your suggestion. In this study, we measured the concentration of total iron in the river water, not iron ions.

Line 220—delete--- figure illustrates

Response: Thanks for your suggestion. We have removed the caption of the explanation and insert in the figure.

Lines 213-218--- more clarity is required here

Response: We have added the explanation of how the model works in line 129-137.

Lines 224-225, It was not clear how this conclusion was decided. However, it would be good to focus the results in view of the PMF Model outcomes. 

Response: We have added the explanation of this conclusion in line 226-229.

Round 2

Reviewer 2 Report

I have recommended rejection of the paper, following your revisions. You have not adequately addressed my previous comments